# A Computational Model of the Respiratory CPG for the Artificial Control of Breathing

**DOI:** 10.3390/bioengineering12111163

**Published:** 2025-10-26

**Authors:** Lorenzo De Toni, Federica Perricone, Lorenzo Tartarini, Giulia Maria Boiani, Stefano Cattini, Luigi Rovati, Dimitri Rodarie, Egidio D’Angelo, Jonathan Mapelli, Daniela Gandolfi

**Affiliations:** 1Department of Biomedical, Metabolic and Neural Sciences, University of Modena and Reggio Emilia, I-41125 Modena, Italy; 242872@studenti.unimore.it (L.D.T.); federica.perricone@unimore.it (F.P.); lorenzo.tartarini@unimore.it (L.T.); 2Institute of Biophysics, National Research Council (CNR), I-90146 Palermo, Italy; giuliaboiani@unimore.it; 3Department of Engineering “Enzo Ferrari”, University of Modena and Reggio Emilia, I-41125 Modena, Italy; stefano.cattini@unimore.it (S.C.); luigi.rovati@unimore.it (L.R.); 4Department of Brain and Behavioral Sciences, University of Pavia, I-27100 Pavia, Italy; d.rodarie@gmail.com (D.R.); egidiougo.dangelo@unipv.it (E.D.); 5Brain Connectivity Center, IRCCS Mondino Foundation, I-27100 Pavia, Italy; 6Center for Neuroscience and Neurotechnology, University of Modena and Reggio Emilia, I-41125 Modena, Italy

**Keywords:** artificial breathing, Computational Neuroscience, respiratory CPG, closed-loop breathing control, neuronal networks, neuromorphic systems

## Abstract

The human respiratory Central Pattern Generator (CPG) is a complex and tightly regulated network of neurons responsible for the automatic rhythm of breathing. Among the brain nuclei involved in respiratory control, excitatory neurons within the PreBotzinger Complex (PreBötC) are both necessary and sufficient for generating this rhythmic activity. Although several models of the PreBötC circuit have been proposed, a comprehensive analysis of network behavior in response to physiologically relevant external inputs remains limited. In this study, we present a computational model of the PreBötC consisting of 1000 excitatory neurons, divided into two functional subgroups: the rhythm-generating population and the pattern-forming population. To enable real-time closed-loop simulations, we employed parallelized multi-process computing to accelerate network simulation. The network, composed of asynchronous neurons, could produce bursting activity at a eupneic breathing frequency of 0.22 Hz, which could also reproduce the rapid and stable chemoreception of breathing activated in response to hypercapnia. Additionally, it successfully replicated rapid and stable respiratory responses to elevated carbon dioxide levels (hypercapnia), mediated through simulated chemoreception. External inputs from a carbon dioxide sensor were used to modulate the network activity, allowing the implementation of a real-time respiratory control system. These results demonstrate that a network of asynchronous, non-bursting neurons can emulate the behavior of the respiratory CPG and its modulation by external stimuli. The proposed model represents a step toward developing a closed-loop controller for breathing regulation.

## 1. Introduction

The Central Pattern Generators (CPGs) are networks of neurons responsible for coordinating stereotyped motor functions such as walking, running, chewing or deglutition [1]. Among these rhythmic activities, breathing is an autonomic process regulated by the respiratory CPG located in the brainstem—primarily within the medulla oblongata and pons [2]. The respiratory CPG orchestrates the motor output that drives the muscles involved in breathing, as well as associated behaviors like swallowing [3] and coughing [4]. The mammalian respiratory CPG has been extensively studied because its output patterns can be observed under various experimental conditions [5,6]. It generates the basic rhythm of breathing, which operates independently of conscious inputs, although it can be modulated by higher brain centers in response to factors such as physical activity, emotional states, or speech [7].

This neural circuitry is evolutionary conserved (Figure 1), and the preBotzinger¨ complex (PreBötC)—a cluster of neurons in the ventrolateral medulla—is considered the core rhythmogenic site. The PreBötC consists of loosely distributed excitatory glutamatergic neurons and inhibitory GABAergic and glycinergic neurons [8,9] and is often referred to as the “pacemaker” for breathing. Neurons in the PreBötC exhibit intrinsic rhythmic firing activity and are essential for respiratory rhythm generation [8,10,11,12]. Damage to this region can significantly disrupt or even abolish normal breathing. However, the exact mechanisms by which the PreBötC generates rhythmic output remain under debate. Current research focuses on cellular and network-level properties that underlie rhythmogenesis [13]. Numerous experimental [12,14,15,16] and computational modeling studies [17,18,19,20,21,22] have explored the cellular and synaptic architecture of this system, leading to two major competing theories: the “pacemaker” and the “burstlet” hypotheses.

The first hypothesis posits that a subset of neurons in the PreBötC continuously generates bursts of action potentials in in vitro preparation [10,23,24], initiating the synchronization necessary for rhythmogenesis. In contrast, the burstlet hypothesis, rooted in the “group pacemaker model, suggests that non-pacemaker neurons interact through excitatory feed-forward synapses, resulting in low synchrony activity (burstlets) that eventually trigger full pre-inspiratory bursts [25,26,27].

Although both theories have received substantial support over the years, a recent computational model [19] proposed a compelling unified hypothesis: synaptically coupled bursting neurons do not produce coherent rhythmic activity, whereas asynchronous non-bursting neurons, when connected by excitatory synapses, can spontaneously generate rhythmic activity. Despite this theoretical insight and supporting evidence suggesting that pacemaker neurons may play a modulatory rather than essential role in the rhythmogenesis [24,28], direct experimental proof that network dynamics alone are necessary and sufficient for autonomous breathing is still lacking. to sustain autonomic breathing has yet to be provided.

Beyond the PreBötC, the breathing cycle is shaped by other clinical brainstem structures. These include (i) the dorsal respiratory group (DRG), which primarily regulates inspiration by processing chemosensory inputs related to blood CO_2_ levels; (ii) the ventral respiratory group (VRG), which contributes to both inspiration and expiration, especially during active or forced breathing [29,30,31]; the pontine respiratory group (PRG), located in the pons, which adjusts the transition between inspiration and expiration, adapting the respiratory pattern in response to behavioral states like exercise or sleep [32]. A crucial structure in this system is the Retrotrapezoid Nucleus (RTN) [7,33], a small but vital area of the brainstem (Figure 1). The RTN plays a central role in respiratory chemoreception, detecting changes in CO_2_ and pH in the cerebrospinal fluid (CSF) and sending feedback to the PreBötC and DRG to modulate breathing rate and depth. When CO_2_ levels rise, pH drops, and RTN neurons increase their firing to stimulate respiratory centers, helping maintain acid–base homeostasis. Dysfunction of the RTN can result in abnormal respiratory patterns, often due to impaired CO_2_ detection and signaling, leading to defective breathing control.

A comprehensive understanding of the mechanisms underlying respiratory rhythmogenesis, including how external inputs modulate CPG activity, remains a significant challenge. Due to the complexity of the intricate neural networks that directly and indirectly influence respiratory rhythm, experimental approaches are often limited to examining only a small subset of components within the circuit. For example, studies investigating the effects of hypoxia on respiratory patterns [34], the mechanisms of CO_2_/H^+^ chemoreception within respiratory circuits [35], or the role of RTN neurons [36] in maintaining CO_2_ homeostasis and controlling breathing, are typically confined to in vitro pharmacological investigation targeting the activity of individual neurons. Given this complexity, there is a growing need for alternative approaches, such as computational modeling, which can replicate key properties of these neural circuits and generate predictions about functional behaviors that are otherwise inaccessible through experimental methods. To shed light on this complex process, we propose a computational approach based on biologically realistic neural models of the PreBötC and its modulation by chemoreceptive inputs from the RTN. We propose a computational model capable of generating and processing biologically relevant stimuli, while preserving the functional architecture and dynamics of the underlying neural circuits.

Moreover, the proposed approach has been applied to a specific clinical scenario: the use of extracorporeal membrane oxygenation (ECMO), the primary therapy for refractory hypoxemic respiratory failures, in combination with mechanical ventilation. In this context, mechanical ventilation must manage both artificial and native lungs, and determining the optimal ventilation strategy remains an active area of research. While standard guidelines typically advocate for lung protective ventilation to minimize further injury, some centers adopt a more physiological ventilation strategy to promote lung recruitment [37,38]. Despite the variety of ventilator management strategies, to our knowledge, none of the current approaches aimed at preserving physiological lung function in acute respiratory distress syndrome (ARDS) patients incorporate biomimetic neuronal networks as a closed-loop controller [39]. In the context of real-time respiratory rate modulation, it is conceivable to develop a controller that mimics the endogenous CPG and integrates physiological feedback inputs [40]. Among the key parameters that reflect the patient’s metabolic demand and influence the respiratory regulation are the level of pCO_2_ [41] and pH [42]. These can be continuously monitored using optical sensors [41,43] and used to control ventilation systems based on artificial neural networks, as recently proposed [44]. There is growing evidence that AI-driven systems can be effectively and reliably used to analyze physiological signals [45,46]. This advancement holds significant potential for the development of brain–computer interface applications, providing valuable tools for both clinical practice and research purposes [47]. Within this wide framework, we validated our approach by interfacing the artificial CPG with real-time data from an optical sensor, the arterial carbon dioxide (PaCO_2_) values extracted from the ECMO circuit. This implementation represents a first step toward a biomimetic closed-loop control system, in which mechanical ventilation could be dynamically regulated based on continuous monitoring of the patient’s blood parameters, thus enabling adaptive artificial respiratory support.

## 2. Materials and Methods

### 2.1. The E-GLIF Neuron Model

The neuron models used to simulate Rhythm and Pattern population in the PreBötC were adapted from [48] and implemented in NESTML (https://nestml.readthedocs.io/en/latest/; accessed on 1 September 2025) as Extended Generalized Leaky Integrate and fire (EGLIF) neuron models [48,49,50,51]. Briefly, each model is defined by a set of parameters, including: external stimulation current (I_stim_), membrane capacitance (C_m_), membrane time constant (τ_m_), resting potential (E_L_), endogenous current (I_e_), adaptation constants (k_adap_, k_2_), decay rate of depolarizing current I_dep_ (k_1_), threshold potential (Vth), escape rate parameters (λ_0_,τ_V_), post-time spike (t_spk+_), refractory period t_ref_, reset potential (V_r_), and current update constants (A_2_, A_1_). To produce asynchronous firing at 0.1 Hz in the rhythm population (Figure 2A), the default model parameters—namely V_m_ (mV), τV (mV), λ_0_ (ms^−1^), k_2_ (ms^−1^), k_adap_ (MH^−1^), k_1_ (ms^−1^), A_2_ (pA), t_ref_ (ms) were adjusted accordingly. In contrast, for the pattern population, V_m_ was initialized to match the resting potential E_L_, resulting in initially silent neurons. Additionally, k_2_ was set to 0.03333 (1/τ_m_) to achieve a non-damping dynamic (see Table 1).

In contrast, for the pattern-generating population, V_m_ was initialized to match the resting potential E_L_, resulting in initially silent neurons. Additionally, k_2_ was set to 0.03333 (i.e., 1/τ_m_) to achieve non-damping dynamics (see Table 1).

### 2.2. The Synapse Model

Synapses were implemented in the NEST simulator [52] (https://www.nest-simulator.org/) as static, conductance-based connections, each assigned specific weights depending on the type of synaptic interaction. These weights were carefully adjusted to elicit tunable bursting that could fit the physiological range of preBötC neurons firing (see Table 2). Synaptic weights were tuned to preserve stable rhythm and burstlet dynamics consistent with the experimentally observed activity ranges.

In particular

Synaptic weights within the rhythm population (*rr*) were set to 9.4.Connections from the rhythm to the pattern population (*rp*) were assigned a weight of 1.0.Connections from the pattern to the rhythm population (*pr*) were set to 0.1.Synapses within the pattern-generating population (*pp*) had a weight of 0.5.Static synapses from the spike generators—ideally representing the RTN—projecting to the rhythm population were set to a weight of 2.9.

All the connections were excitatory and attached to a receptor type 1 (glutamatergic AMPA).

### 2.3. The Network Model

The network model was implemented using the NEST simulator (v3.7), with simulations executed at a temporal resolution of 0.1 ms via the PyNEST (v3.10) and PyNESTML(v7.0) interfaces [53]. Based on recent experimental and computational findings [12,20], demonstrating that rhythmogenesis can occur independently of inhibitory interneuron activity, the PreBötC was modeled as a network of 1000 excitatory neurons, divided into two functionally distinct subpopulations: Rhythm and Pattern. The Rhythm population (250 neurons) consists of pre-inspiratory neurons responsible for rhythm generation. These neurons exhibit firing behavior compatible to that of somatostatin-negative (SST−) neurons identified in animal models [14,15]. At the onset of each respiratory cycle, these neurons fire asynchronously and spontaneously, progressively synchronizing during the interburst interval [15]. This synchronization gives rise to burstlets, which are considered the rhythmogenic signature underlying the initiation of the breathing cycle [12,26,54]. In contrast, the pattern population (750 neurons) is responsible for shaping the respiratory output, mirroring the functional role of somatostatin-positive (SST+) neurons. These neurons are recruited when the Rhythm population achieves sufficient synchrony and, in turn, drive the onset of inspiratory motor activity. The RTN was modeled as a network of 120 neurons modeled using NEST’s spike generator devices. Each generator was configured to emit bursts of three spikes at an intra-burst frequency of 100 Hz. These bursts were repeated at fixed intervals, thereby defining the frequency at which the RTN modulates PreBötC activity. Connectivity within and between populations was established using the “pairwise Bernoulli” rule, where connections between source and target nodes are probabilistically generated based on a Bernoulli process. The intra-population connection probabilities were set at 13% for the rhythm population and 2% for the pattern population in accordance with experimental observations [15,55]. Bidirectional connections between the rhythm and pattern populations were assigned a 30% connection probability, ensuring that the overall connection density within the PreBötC network remained approximately 13% [20,55]. The RTN-to-Rhythm population connectivity was set to 17%, while no connections were made from the RTN to the pattern population, reflecting the selective targeting of rhythm-generating neurons by chemoreceptive input (see Table 2).

### 2.4. The Respiratory Rate Model

The modulation of the CPG network by the RTN was conceived to be compliant with the pH compensation process, which is the mechanism adopted by the body when there are changes in blood pH, such as in the case of acidosis or alkalosis, and, therefore, attempts to restore the acid–base balance. Central to this process is the buffering system (bicarbonate), as it helps to understand how the respiratory and renal systems work together to maintain the pH balance. When one system is unable to compensate, the other will step in to restore balance. The respiratory system is more rapid as it adjusts the pH by regulating CO_2_ levels within minutes to hours, while the renal system has a slower response but can keep the pH balance over a longer duration by adjusting the excretion of hydrogen ions and bicarbonate. The buffering system is described by the Henderson–Hasselbalch equation [54].(1)$$pH=p_{Ka}+\log_{10}\frac{\left[{HCO}_{3}^{-}\right]}{\alpha \times {pCO}_{2}}$$
where [HCO_3_^−^] is expressed in mmol/L and the pCO_2_ in mmHg. The constant pK_a_ represents the negative logarithm of the acid dissociation constant (K_a_) of a weak acid. It is a measure of the strength of the acid, indicating the pH at which the acid is 50% dissociated into its conjugated base and proton (H^+^), and for carbon acid it is 6.1.

When the pH in the blood shifts from the normal range (around 7.4), the body activates compensatory mechanisms to bring it back to the optimal value. In practice, Equation (1) highlights how blood pH depends on the ratio between the concentration of bicarbonate [HCO_3_^−^], which is regulated by the kidneys, and the amount of CO_2_ in the blood, which in turn depends on pulmonary ventilation (breathing).

According to Henry’s law, the concentration of a dissolved gas in a liquid is directly proportional to the partial pressure of the gas and its solubility coefficient. In Equation (1), α is the solubility coefficient of CO_2_ in blood plasma at 37 °C. This value is 0.07 mL CO_2_/(100 mL of blood per mmHg), which corresponds to an increase in concentration of approximately 0.03 mmol/L for each mmHg of pressure increase (Table 3). By rearranging (1), we can derive the expression for pCO_2_ as a function of the bicarbonate concentration [HCO_3_^−^] and pH:(2)$${pCO}_{2}=\frac{\left[{HCO}_{3}^{-}\right]}{0.03\times 10^{pH-6.1}}\;$$

This expression allows us to calculate the pCO_2_ value required to compensate for changes in bicarbonate concentration in response to metabolic acidosis or alkalosis, assuming a constant pH of 7.4. We can also consider the relationship between the alveolar ventilation (${\dot{V}}_{A}$) and the arterial pCO_2_ (P_a_CO_2_), described by:(3)$${\dot{V}}_{A}=\frac{0.863\times \dot{V}{CO}_{2}}{P_{a}{CO}_{2}}$$
where $\dot{V}{CO}_{2}$ is the carbon dioxide production (mL/min), and the constant 0.863 is included to convert units of $\dot{V}{CO}_{2}($mL/min) and ${\dot{V}}_{A}$ (L/min) to units of pCO_2_ (mmHg). In the approximate treatment of alveolo-capillary equilibrium, we assume that pCO_2_ = PaCO_2_, allowing us to express (2) in terms of PaCO_2_. Additionally, we can introduce the relationship that connects the alveolar ventilation ${\dot{V}}_{A}$ (L/min) and the alveolar volume $V_{A}$ (L) with the number of breaths per minute (bpm):(4)$$bpm=\frac{{\dot{V}}_{A}}{V_{A}}$$

Alveolar volume ($V_{A}$) is the difference between tidal volume, the volume of air entering the airways at each breath, and the volume of anatomical dead space, the part of the airways that does not allow gas exchange. From this, we can determine the time interval between two consecutive breaths by dividing one minute (60,000 ms) by the bpm:(5)$$interval=\frac{60,000}{bpm}$$

Similarly, we can express the associated respiratory rate frequency in Hz:(6)$$f=\frac{bpm}{60}$$

In the present study, we investigated the response of the PreBötC to the modulatory inputs of the RTN induced by variations in the carbon dioxide production ($\dot{V}{CO}_{2}$) and bicarbonate concentration [HCO_3_^−^]. The activity of the RTN was calibrated to model changes in $\dot{V}{CO}_{2}$, under the conditions of constant physiological values of pH, [HCO_3_^−^], PaCO_2_ and $V_{A}$. The $\dot{V}{CO}_{2}$ was varied between 250 mL/min and 620 mL/min, and the corresponding alveolar ventilation was computed by using (3). The number of breaths per minute and the corresponding interval for each $\dot{V}{CO}_{2}$ value was then calculated by using (4) and (5). The intervals were adopted to configure the stimulation frequency of RTN. The previous equations can be condensed into a single equation that describes the dependency of RTN frequency on the variation in $\dot{V}{CO}_{2}$:(7)$$f_{RTN}\left(\dot{V}{CO}_{2}\right)=\frac{0.863\times \dot{V}{CO}_{2}}{40.1\times 0.35\times 60}=c_{1}\times \dot{V}{CO}_{2}$$

All fixed parameters of Equation (7) can be incorporated in a single constant c_1_, highlighting the linear dependence of RTN frequency with respect to $\dot{V}{CO}_{2}$. Additionally, we analyzed how the RTN activity is influenced by variations in [HCO_3_^−^], which varied from 10 mmol/L to 24 mmol/L, corresponding, respectively, to acidosis and alkalosis conditions. For each [HCO_3_^−^] value, using (2) and assuming a constant pH of 7.4, we calculated the PaCO_2_ value required to achieve a compensated system. While keeping $\dot{V}{CO}_{2}$ constant at its physiological value, the calculated PaCO_2_ value was substituted into (3) to obtain the alveolar ventilation for each [HCO_3_^−^] value. Next, by inserting the determined ${\dot{V}}_{A}$ value into (4), and assuming a constant alveolar volume $V_{A}$, we obtained the bpm and the corresponding interval through Equation (6). These steps can be synthesized into a single equation that describes the variation in RTN frequency as [HCO3^−^] changes:(8)$$f_{RTN}\left(\left[{HCO}_{3}^{-}\right]\right)=\frac{0.863\times 250\times 0.03\times 10^{pH-6.1}}{\left[{HCO}_{3}^{-}\right]\times 0.35\times 60}$$(9)$$=\frac{c_{2}}{\left[{HCO}_{3}^{-}\right]}$$

### 2.5. Burstlet–Burst Analysis

Bursts and burstlet detection was performed by analyzing the interspike interval (ISI) within neuronal spike trains. For the rhythm population, a neuronal burst was defined as the occurrence of at least three consecutive spikes separated by an ISI of less than 40 ms. A population-level burst was identified when all rhythm neurons exhibited a burst within a 250 ms time window. Conversely, a burstlet was defined as a partial synchronization event in which fewer than 40% of rhythm neurons emitted a burst within the same 250 ms time window. Burstlets were typically subthreshold events that failed to activate the pattern population. Within the pattern population, a neuronal burst was defined by the presence of at least two consecutive spikes separated by an ISI shorter than 100 ms. Burst formation in this population exhibited higher variability, with over 87% of neurons participating in each detected burst. The intraburst firing frequency was computed as the inverse of the ISI (1/ISI), with ISI values converted to seconds to express frequency in hertz (Hz). To assess the temporal synchronization during bursts, a sliding window analysis was performed. For pattern burst, a 50 ms sliding window was applied from burst onset to offset, corresponding to an average firing frequency of 20 Hz. In each window, the number of active neurons was counted and normalized to the total population, yielding the percentage of active neurons over time. The same procedure was applied to the Rhythm bursts, but with a shorter window of 20 ms, reflecting their higher average firing frequency of approximately 50 Hz.

### 2.6. Interburst Variability

To assess interburst variability, we employed two complementary measures: rhythm synchronization and burst onset latency. Rhythm synchronization quantifies the degree of neuronal coordination within the Rhythm population during the interburst interval, while burst onset latency measures the temporal delay between rhythm generation and the activation of the Pattern population. To calculate rhythm synchronization, a 20 ms sliding window with a 10 ms step was applied across the activity of the Rhythm neurons. The onset of synchronization was defined as the first window in which at least 10% of Rhythm neurons were active. Rhythm synchronization was then quantified as the time interval between this detected onset and the subsequent population burst within the Rhythm population. In parallel, burst onset latency was computed as the time difference between the Rhythm burst onset and the initiation of a corresponding burst in the Pattern population, thereby capturing the delay required to recruit the downstream network.

### 2.7. Network Performance

The network performance was evaluated through a four-minute simulation to assess the stability and persistence of activity over extended periods. In addition, we investigated the influence of stochastic variability by repeating simulations with different seeds used to initialize random processes, which determined connectivity patterns and spike generation. In total, fourteen independent simulations of one minute each were conducted, each with a unique seed. To test the scalability of the model, we also performed a one-minute simulation using a larger network of 4000 neurons.

We further examined the robustness of network activity under alternative connectivity configurations. Specifically, we varied the connection probability between Rhythm and pattern populations, while maintaining the experimentally observed connection probabilities within populations (13% for Rhythm-Rhythm at 2% for Pattern-Pattern connections) [15,55]. Bidirectional Rhythm-Pattern connectivity was first increased to 40% and then decreased to 20% and 13%. We also tested asymmetric configurations, setting the Rhythm-to-Pattern connection probability at 40% and later reducing it to 13%, while keeping the Pattern-to-Rhythm connection probability fixed at 2%. To preserve the overall excitability of the network across these varying topologies, synaptic weights are adjusted using a normalization parameter J, defined as J = P_i_n_i_*w_i_ where i denotes the type of connection, n*_i_*is the number of connections, and w*_i_*is the connection weight. Under the default configuration, the reference value of J was approximately 62,000. In all tested configurations, synaptic weights were scaled to maintain J within a functional range (60,000 to 80,000). This normalization strategy successfully preserved rhythmic activity across different network architectures. Although the main objective of this study is the development of a real-time closed-loop system for biomimetic respiratory control, executing neuronal simulations in real time on conventional computing platforms presents a significant challenge. In our benchmark, a 60,000 ms simulation took approximately 8 min, while 120,000 ms simulations took 16–17 min. The four-minute (240,000 ms) simulation completed in approximately 34 min. When RTN input was included, was included, simulations of 80,000 ms, 120,000 ms, and 180,000 ms required 12–13 min, 20 min, and 30 min, respectively. The scale-up simulation with 4000 neurons required 20 min to be completed. Real-time communication was achieved by optimizing both hardware and software performance. Parallel, multi-process computation allowed for the simulation of 1 min of activity in a network of 1000 E-GLIF neurons in less than 45 s, thereby demonstrating the feasibility of real-time closed-loop respiratory control. This can be possible by a communication protocol capable of simulating temporal blocks and updating digital I/O signals in a quasi-continuous manner. Additional sources of I/O latency, including burst detection and conversion of pCO_2_ values into spike trains, were also optimized. Both operations were completed in less than 30 microseconds, leaving ample temporal margin for dynamic regulation of breathing, which operates on a second timescale. As a result, transitions between different respiratory rates depend solely on pCO_2_ sampling frequency, with breathing adjustments occurring between consecutive respiratory acts (see Figure 2 in [40]). Finally, scaling up the network to include additional neuronal populations involved in modulatory functions will require enhancements in the simulation infrastructure. The increased computational demands may be addressed by leveraging accelerated hardware like GPUs [56] or alternatively adopting neuromorphic computing platforms, including commercial devices (e.g., Loihi, Dynap-se or Akida [57,58,59]) or custom-designed neuromorphic architectures [60,61].

### 2.8. Real-Time Application for Artificial Breathing

The network model was integrated with a real-time PaCO_2_ sensor developed in [62], which continuously measures the arterial partial pressure of carbon dioxide on a blood line. To dynamically modulate network activity, the recorded PaCO_2_ value was converted into an RTN interval firing frequency to subsequently update the RTN interval every 3 s. At each update, the interval was calculated based on the relationship described by [63], applying the breathing frequency parameters (T_1_, S_1_, T_2_ and S_2_) extracted from Table 1 in [63] for a PO_2_ value of 80 mmHg. For PaCO_2_ values lower than 44.3 mmHg, the RTN was silenced to maintain the basal respiratory rate of 13 bpm (see Table 4). For PaCO_2_ values between 44.3 and 52.2 mmHg, corresponding to T_1_ and T_2_ points, the relationship between PaCO_2_ and the ventilatory response was assumed to be linear with a slope S_1_ of 0.7. The corresponding equation, *RR*_1_ = 0.7 × *PaCO*_2_ − 18.01, was derived by applying the equation of a straight line, using T1 as the x0 point and setting y0 as the basal respiratory rate of 13 bpm. The corresponding interval was then calculated as the inverse of RR_1_ multiplied by 60,000. For PaCO_2_ values exceeding 52.2 mmHg, the slope S_2_ of 1.5 was applied, resulting in the following equation: *RR*_2_ = 1.5 × *PaCO*_2_ − 59.77. This equation was derived by applying the equation of a straight line, selecting T_2_ as the x_0_ point and using y_0_ as the value of RR_1_ calculated at T_2_. Again, the corresponding interval was derived as the inverse of RR_2_ multiplied by 60,000. The PaCO_2_ data used in the simulation, taken from [63], ranged from 36.8 mmHg to 61.0 mmHg (Table 4). To preserve the state of the network between updates, the simulation was implemented as a single continuous run in NEST, achieved by concatenating multiple 3 s runs without restarting the simulation kernel. This setup enabled the generation of respiratory activity capable of compensating in real time for an out-of-equilibrium condition continuously monitored by the system.

### 2.9. Statistical Analysis

Data are reported as means ± standard error of the mean (SEM) unless otherwise stated.

### 2.10. Data and Code Availability

Data and code reported in this paper will be shared by the lead contact upon request. Any additional information required to reanalyze the data reported in this paper is available from the lead contact upon request.

## 3. Results

The respiratory CPG was modeled as a network of neurons that achieved collective synchronization through specific connectivity patterns. This circuit, representing a simplified PreBötC, consisted of 1000 point-neurons organized into two subpopulations—Rhythm and Pattern—interconnected both internally and reciprocally via purely excitatory synapses [20] (see Section 2).

Due to the limited availability of electrophysiological data on single neurons and microcircuits in the human central nervous system, we have hypothesized that, since the respiratory CPG is a highly conserved structure across mammalian species, human circuits could reflect the functional and structural organization of rodents. Accordingly, we implemented a neuronal circuit reflecting rodents’ architectures to be upscaled to reproduce the functional dynamics observed in humans. Given these premises, neurons in both populations were modeled using a Generalized Leaky Integrate-and-Fire (E-GLIF) neuron model (see Section 2.1). This model incorporates spike-triggered currents and a dynamic spike-triggered threshold [50], allowing it to replicate a wide range of electrophysiological behaviors while maintaining biological plausibility and computational efficiency. Notably, the model supports multiple dynamic regimes depending on parametric tuning (see Figure 2A in [48]). To reproduce the spontaneous firing observed in Rhythm Population neurons, we adjusted the parameters k_2_ and k*_adapt_*—which govern adaptation dynamics—to generate a stable yet asynchronous firing at approximately 0.1 Hz (see Appendix A). In line with experimental findings [64] and recent computational models [20], the 250 neurons in the rhythm population were connected (see Section 2) with a 13% connection probability. This configuration resulted in an asynchronous firing across individual neurons that gradually converged to a quasi-stable bursting pattern. This activity was characterized by high frequency intra-burst firing (50.0 ± 0.5 Hz) and low frequency inter-burst intervals (0.22 ± 0.01 Hz), consistent with the eupneic respiration rhythm of adult humans (0.20–0.33 Hz; see Figure 3A). To assess construct validity, we systematically varied the connection probability within the Rhythm population. Bursting activity was silenced when connectivity was lowered below 9% while values above 17% generated an almost continuous firing (see Appendix A). In parallel, according to experimental findings [15], neurons in the pattern population were sparsely and randomly connected with a 2% connection probability [15], resulting in near-silent activity. The rhythm and pattern populations were then reciprocally connected with 30% excitatory synapses. As the Rhythm population generated increasingly synchronized activity (burstlets), neurons in the Pattern population were progressively recruited, culminating in robust bursting activity (Figure 3B). This activity in biology propagates through the rVRG pathway and the phrenic nerve to induce a motor output. Is should be noted that connection probabilities for the Rhythm [61] and pattern [15] populations were calibrated using experimental data, while the reciprocal connectivity between them was adjusted within a range that maintained the overall 13% connectivity observed in the PreBötC [61] since no experimental data are available from the literature. Strikingly, the pattern population exhibited inter-burst frequencies matching those of the Rhythm population (0.22 Hz; Figure 3B), while the intra-burst frequencies were significantly lower (22.0 ± 0.3 Hz, SEM). Since our model relies on single-neuron adaptation dynamics (see inset Figure 3), which in the E-GLIF model are stochastic, the network’s inter-burst activity varied from cycle to cycle [15,65]. To quantify synchronization, we computed the temporal delays between successive bursts in the Rhythm population and the corresponding onset times in the Pattern population (see Section 2.5 and Section 2.6). Over 120 s of simulation, the average burst synchronization time in the Rhythm population was 821.1 ± 232.1 ms (ranging from 249.6 ms to 4620.5 ms). The mean burst onset delay in the Pattern population was 43.7 ± 1.1 ms (ranging from 32.6 ms to 54.4 ms). Burst synchronization analysis showed that in the first two burstsv96.4% and 90.0% of Pattern neurons, respectively, fired at least once. Only 2 of the 25 bursts involved fewer than 90% of active neurons. In the Rhythm population, four bursts involved more than 90% of active neurons, with 13 bursts reaching full anticipation (100%). In contrast, burstlets—defined as synchronized events involving fewer than 30% of neurons (Figure 3C, red arrow)—failed to recruit the Pattern population and did not produce motor bursts.

To introduce variability into the network structure, we performed multiple simulations using different random seeds for (Figure 3D). Across 14 simulations, temporal variability was quantified: bursts in the rhythm population emerged after 3076.8 ± 9.9 ms (range: 2958.7–3404.4 ms). We also assessed variability in the onset time of the first burst, which exhibited a mean value of 26.7 ± 0.3 ms (range: 21.5–33.9 ms). The average inter-burst frequency was 0.21 ± 0.02 Hz (mean ± SD), while intra-burst rates were 49.5± 0.23 Hz in the Rhythm population and 21.9 ± 0.25 Hz in the Pattern population. Importantly, bursting activity in the Rhythm population remained stable over time, with minimal variability attributable to slight fluctuations in spike timing and synchronization. In simulations where the Rhythm population failed to synchronize effectively, the Pattern population generated an output (red arrows in Figure 3C,D). However, bursting activity was restored in subsequent cycles. This behavior aligns with experimental observations [66] and supports the hypothesis that synchronization governs the transition between full bursts and burstlets—shorter, subthreshold synchronous events that do not recruit the Pattern population. This mechanism enables finer modulation of respiratory rhythms, offering a degree of tunability based on network dynamics. When synchronization fails to reach a critical threshold, burstlets are terminated, and the network reverts to a desynchronized state. The RTN plays a crucial role in maintaining blood gas homeostasis. This population of glutamatergic neurons responds robustly to elevated arterial CO_2_ (pCO_2_) levels [33], and their essential role in central chemoception is demonstrated by the shift in CO_2_ sensitivity observed when RTN neurons are silenced [67,68]. RTN neurons detect changes in pH [69], remaining quiescent under alkaline conditions and discharging in response to acidification, with an approximate firing rate increase of 0.5 Hz per 0.01-unit decrease in pH (see Table 4) [69]. Thus, they are well-positioned to mediate respiratory compensation in response to interstitial pH variations. RTN neurons were incorporated into our CPG model, and their activity was simulated according to pH-driven respiratory modulation (see Section 2). The main target of the present work is to provide proof of concept that spiking networks can be used to mimic the autonomous breathing in the preBötC and can be modulated by external inputs delivered through. The current version is an intermediate stage, allowing the conversion of pCO_2_ input into spikes. RTN neurons were, therefore, modeled as simple excitatory-only elements connected exclusively to the Rhythm population of the preBötC, with a connection probability of 17%, optimized to ensure stable and effective coupling. Reducing this connectivity below 15% disrupted communication between RTN and preBötC.

The number of neuronal elements in the RTN model is significantly lower than the biological counterpart (120 neurons vs. about 2000 neurons). To increase the efficiency of spike transmission, preserving the connection probability, we could either increase synaptic weights or spike frequency (Appendix A). We have, therefore, decided to generate RTN activity using spike generators emitting 100 Hz spike trains, repeated at variable frequencies (Figure 4A). The inter-train interval was calibrated to reflect physiological variations in carbon dioxide production ($\dot{V}{CO}_{2}$) and [HCO_3_^−^] while keeping constant pH, pCO_2,_ and VA (see Section 2). In the presence of RTN input, the firing frequency of the PreBötC Rhythm population increased slightly and stabilized within physiological limits (green square in Figure 4A). Higher stimulation frequencies further enhanced synchronization in both Rhythm and Pattern populations. These initial simulations confirmed that PreBötC rhythmogenesis is susceptible to modulation by external stimuli from the RTN. When RTN trains were delivered at 0.25 Hz (physiological condition) and 0.44 Hz (hypercapnic condition) (see Section 2 and Equation (1)), the Rhythm population synchronized accordingly, and this activity was transmitted to the Pattern population. In a subsequent set of simulations, we explored the system’s stability under external perturbations by varying RTN firing frequencies to reflect physiological [HCO_3_^−^] concentrations derived from Equation (1). Based on theoretical predictions (Equation (5)), spike trains were generated across a range of 0.26 Hz (corresponding to 24 mEq/L [HCO_3_^−^]) to 0.62 Hz (corresponding to 10 mEq/L [HCO_3_^−^]) (Figure 4B). Similarly, experimental findings in animal models have shown that in conditions of severe hypercapnia, changes in RTN and PreBötC firing frequencies reach 2.5 times the physiological value [70,71]. RTN stimulation effectively entrained the Pattern population, leading to synchronized bursting at frequencies matching the input. Remarkably, experimental results from repeated simulations closely followed the theoretical hyperbolic relationship between respiratory frequency and [HCO_3_^−^] (Equation (8)). Finally, we calibrated RTN activity based on changes in $\dot{V}{CO}_{2}$ due to varying CO_2_ flux, as modeled by Equation (7). As expected, the artificial CPG responded robustly to CO_2_-driven ventilation signals (Figure 4C), demonstrating that the model accurately reproduces respiratory rhythmogenesis and its modulation by chemosensory feedback.

Critically, the artificial CPG system—comprising the PreBötC and RTN circuitry—demonstrated the ability to adapt to complex and continuous dynamic perturbations, as observed during physiological environmental fluctuations (Figure 5). This computational model shows strong potential for integration with real-time physiological monitoring systems and mechanical ventilation devices in the context of critical care in critical patients. Specifically, the model’s adaptability was tested by modulating the activity of chemosensitive RTN neurons in response to arterial carbon dioxide partial pressure (PaCO_2_) values, as measured by a recently developed sensor [62]. PaCO_2_ levels were sampled from the bloodline of an extracorporeal membrane oxygenation system (ECMO) and ranged from 36.8 to 61 mmHg. RTN frequencies were adjusted based on the experimentally derived relationship between respiration rate and PaCO_2_, as reported in [63] for this physiological range (see Section 2).

In response to PaCO_2_ fluctuations, adjustments in RTN firing frequency successfully modulated respiratory rhythmogenesis governed by PreBötC neuronal populations, thereby enabling real-time control of artificial changes that were able to accommodate the breathing rhythmogenesis controlled by the neuronal, neuronally driven CPG (Figure 6A,B). The simulated respiratory frequencies range from 12 to 40 bpm (0.2–0.65 Hz), which is consistent with the normal range observed in adults (12–20 bpm) [72] and with elevated frequencies under hypercapnic conditions (25–35 bpm). The model reproduces the chemoreflex response to increasing PaCO_2_, with slopes of 0.7 bpm/mmHg for PaCO_2_ between 44.3 and 52.2 mmHg, and 1.5 bpm/mmHg for 1.5 bpm/mmHg for PaCO_2_ between 52.2 and 61 mmHg, matching the experimental values reported in [63] at an iso-oxic PO_2_ of 80 mmHg. These results demonstrate that the model quantitatively captures the expected physiological response to CO_2_, providing indirect validation against published data, even in the absence of direct in vivo or patient-derived measurements.

In the context of real-time closed-loop control of artificial respiration, we benchmarked a feedback mechanism based on both the current network state and sensor input. Within this loop, real-time operation is defined as the difference between the resolution of the simulation, the sensor sampling frequency, and the computational processing time. In the closed-loop system, the neural network drives the ventilatory apparatus by generating rhythmic bursts, while concurrently receiving modulatory input from a pCO_2_ sensor. This input is converted into spike trains that are transmitted via the RTN pathway. Real-time communication is ensured through software and hardware optimizations. Specifically, a network of 1000 E-GLIF neurons running on a conventional CPU can simulate one minute of CPG activity in less than 45 s, thanks to parallel computing strategies. In a clinical setting, this implementation would result in no appreciable delay between actual and simulated time. Moreover, burst detection and the transformation of pCO_2_ measurements into stimulation frequencies are computed in less than 30 microseconds, thereby minimizing communication latency (see Figure 2 in [40]). Consequently, transitions between different respiratory states are constrained only by the sampling rate of the pCO_2_ sensor.

## 4. Discussion

### 4.1. Modeling the Respiratory CPG

In this work, we developed a computational model of the respiratory CPG incorporating the PreBötC as the rhythm-generating core and the RTN as a modulator that adjusts respiratory frequency in response to changes in blood gas parameters. The inclusion of the RTN is particularly valuable, as it enables a direct link between the physiological state of the respiratory system and the activity of a specific neuronal population. Computational models of neuronal circuits [73,74] provide a powerful framework for investigating conditions that are often difficult or impossible to probe experimentally [75,76,77]. The respiratory CPG, in particular, poses substantial challenges due to its anatomical location, functional complexity, and limited accessibility, even in small animal models. Previous modeling efforts have largely focused on the PreBötC and its intrinsic mechanisms of rhythm generation. Central to this debate are two major hypotheses: the “pacemaker” and the “burstlets” theories [78]. Although both have been widely studied, the current consensus favors the burstlet hypothesis, which posits that single-neuron autonomous bursting is insufficient for rhythm generation. Instead, synchronous activity among a subset of interconnected PreBötC neurons is required to produce network—level rhythm output. Our modeling approach supports this view, and yields results consistent with experimental data [15] and more biophysically detailed models based on Hodgkin and Huxley neurons [19,79,80]. By employing mono-compartmental integrate and fire neurons, we significantly reduce computational demands, making it feasible to scale the model to include additional neuronal populations involved in respiratory controls [7]. Despite recent insights into cellular and sub-cellular mechanisms underlying the transition from burstlet to full burst, like intracellular Ca^2+^ dynamics [71], persistent sodium current [19], or extracellular potassium accumulation [20], the E-GLIF neuron model effectively reproduces the full range of PreBötC electrophysiological behaviors, including regular spiking, bursting, and tonic firing. These behaviors are achieved by tuning only three key parameters (k_2_ and k_adap_, A_1_), highlighting the model’s simplicity and versatility. The burstlet theory, grounded in experimental observations, proposes that rhythmogenesis arises from a percolation-like process in which neuronal activity gradually synchronizes, culminating in a burstlet. When this activity exceeds a threshold, it recruits the Pattern-generating population, resulting in an all-or-none output [26,27,81]. The present model is consistent with this framework, reproducing a wide range of physiological dynamics and exhibiting robustness to perturbations and firing rate variability. The E-GLIF model is particularly well-suited for this task, as it can transition from tonic, adapting, and bursting firing modes by simply tuning on K_2_ and K_adap_, capturing the diversity of firing behaviors observed in the PreBötC. Moreover, the model aligns with experimental evidence showing that the presence of respiratory rhythmogenesis does not rely on inhibitory activity [82,83]. Furthermore, the role of the inhibitory circuitry in the PreBötC is still under debate, with experimental evidence and computational models showing opposite roles for inhibition. On one side, it is believed to synchronize and regularize bursting [9,84]; on the other side, there is evidence showing a decrease in network synchronization proportional to the level of inhibition [15,85]. We have, therefore, decided not to include, at this stage, the inhibitory circuit. The main purpose of the present work was, in fact, to model a stable respiratory rhythm through a spiking network inspired by the PreBötC architecture. Simulations also support theoretical predictions that rhythm arises from the emergence of synchronized activity—i.e., burstlets—within the PreBötC. This synchronization is not driven by pacemaker neurons but rather emerges from the intrinsic topology and excitatory connectivity among a subset of non-pacemaker neurons [26]. Finally, consistent with these predictions, our simulations demonstrate that voltage-dependent bursting in pacemaker neurons is not required for rhythm generation. Conversely, the presence of a limited number of intrinsically bursting neurons (up to 30% of the rhythm population) as observed biologically, does not disrupt the rhythmic activity (see Appendix A).

In rare cases, burst propagation from the rhythm to the pattern population (see Figure 4) fails, and these failures are incompatible with a stable respiratory activity. In biological circuits, the presence of inhibitory neurons may play a role by sharpening the time window [86,87], increasing synchronization, and reducing the likelihood of motor output failure, as occurring in other brain areas like the cerebellum [88,89,90] or neocortex [91]. Furthermore, since electrical synapses are distributed within the excitatory population with a connectivity rate like chemical synapses [64], it is reasonable to assume that they contribute to synchronization by amplifying excitatory chemical EPSPs. The incorporation of both electrical and inhibitory synapses into the model could, thus, further improve burst reliability and temporal precision. Importantly, the model demonstrated robustness to changes in network connectivity, as long as the parameters remained within physiological limits. When significant architectural changes were introduced, the system’s behavior could be restored through adaptive modulation of synaptic weights. Computational models of the respiratory system have been proposed in the form of biologically realistic [19,69] or integrate and fire neurons [92], and activity-based models [93]. Among these, activity-based models are widely employed due to their rich capacity to represent the diverse set of respiratory nuclei as oscillatory systems. Nevertheless, these models are criticized for their scarce tunability, as the small number of free parameters restricts their adaptability to variable temporal dynamics. More recently, a computational model of the PreBötC has been shown to adapt its rhythmogenesis according to environmental changes such as neurodevelopment, hypoxia, or temperature variations [19]. Different from our model, changes were induced by acting on single neuron parameters like conductance, reversal potentials, or ion concentrations. In contrast, we propose an alternative strategy to mimic central chemoreception by acting at the network level rather than on single neuron parameters. This network-based modulation more closely resembles the biological circuit and provides a realistic and, most importantly, reliable mechanism for respiratory control. Although the PreBötC has been identified in humans [2,8], the precise number and composition of neurons within its subpopulations remain uncertain. Due to the limited availability of human data, we assumed that network architecture could be extrapolated from animal models to humans, based on the evolutionary conservation of the respiratory CPG across species. Accordingly, we maintained the relative size of neuronal subpopulations and their connectivity ratios in the upscaled model. Similarly, the RTN has been characterized in rodents and non-human primates using genetic and molecular markers [33,94], and ongoing studies are working to identify analogous central chemoreceptor neurons in humans [95,96,97].

Since RTN has not yet been identified in humans, its activity has been modeled with a fixed burst (3 spikes at 100 Hz) repeated at a variable frequency, assuming that neurons could behave as in rodents [98], where CO_2_ decrease induces high-frequency bursting in glutamatergic neurons. Future developments of this model will incorporate a more realistic RTN circuit with glutamatergic and GABAergic neurons to better represent a more physiological chemoreception. The RTN in rodents consists, in fact, of a bilateral cluster of approximately 1000 non-catecholaminergic glutamatergic neurons, with around 12% serving as central chemoreceptors [99], which have been the object of our model. Finally, when the model was scaled to match the estimated size of the human PreBötC, the core physiological behaviors were preserved. This suggests that the proposed mechanisms are robust and extendable to include additional nuclei.

### 4.2. Closed-Loop Control of Artificial Respiration

As a preliminary proof of concept, the artificial neuronal circuit demonstrated the ability to autonomously and physiologically regulate respiratory frequency in response to a physiological signal acquired from an external biomedical sensor. The transition from mechanical to physiological respiration represents a major clinical challenge, particularly during the weaning phase, where parameters such as air flow and blood gas levels are critical for monitoring. In clinical practice, it is not possible to define a universal protocol for mechanical ventilation settings during weaning. Instead, it is widely recognized that modulation of respiratory support based on real-time physiological monitoring may enhance the personalization and efficacy of weaning algorithms [100]. Preclinical studies have shown that, in the case of spinal cord injuries [101] or to restore normocapnia after anesthesia [102], the use of bioinspired neural networks can enable physiological regulation of breathing. Similarly, spiking neural networks (SNNs) have been employed to model motor CPG, enabling direct control of robotic actuators via low-power neuromorphic systems for autonomous motor regulation [103,104]. Closed-loop strategies for respiratory control have mainly focused on automated mechanical ventilation guided by diaphragmatic stimulation, either through an external controller [101] or via neuromorphic implants [105]. Our proposed solution aligns with these clinical needs, aiming to implement an artificial CPG capable of modulating ventilation in real-time based on continuous physiological feedback. This approach has the potential to optimize ventilatory support during acute phases but could also enable patient-specific interventions to restore endogenous respiratory functions during recovery. In weaning procedures, where artificial and native lungs work in parallel, the ability of a CPG to drive ventilation according to continuous physiological inputs could mark a paradigm shift and improve the recovery from ECMO. Besides the optimization of artificial mechanical ventilation, a more adaptive, patient-specific intervention could be implemented. The variability of physiological parameters among patients is extremely high, and, despite a series of successes in clinical application of AI-driven [45,46,47] systems, in most cases, the training of the algorithm could be severely limited. The use of SNN driven by physiological inputs could bypass the use of a universal pretrained algorithm in artificial systems. Within this complex framework, the development of integrated hardware solutions is essential. Mixed analog/digital circuits designed to support SNNs have already been proposed, and dedicated neuromorphic platforms have been implemented to simulate the coupling of breathing and heart rate [106], the three-phase respiratory network and the heart chambers’ rhythm [107], and the respiratory motoneurons [108]. The proposed SNN based on E-GLIF could be efficiently translated into neuromorphic hardware using MOSFET [109] or FPGA-based [110] implementation, thereby advancing the concept of bioelectronic medicine. Migrating entirely to hardware solutions would enable closed-loop, real-time operation, already achievable in the current configuration, but could be severely compromised when upscaling the size of the circuit to include millions of neurons. Incorporating additional neuronal populations will demand improvements in simulation efficiency, potentially requiring high-performance hardware accelerators like GPUs [56], commercial neuromorphic devices [57,58,59], or custom neuromorphic architectures [60,61].

## 5. Conclusions

The proposed system holds significant potential in the context of intelligent weaning, where the transition from mechanical ventilation and ECMO to spontaneous physiological breathing could be managed by a bioinspired neuronal network emulating the activity of the respiratory CPG under the control of real-time physiological parameters. The ability to dynamically predict and adapt to a patient’s respiratory capacity represents an intelligent solution that continuously adjusts its output to meet individual metabolic demands. We demonstrate that a spiking model inspired by respiratory CPG closely mirrors natural respiratory behaviors and, if properly developed, could offer clinical support in cases of impaired or absent autonomous breathing. This would be achieved through the implementation of a closed-loop negative feedback control system, enabling the real-time optimization of ventilation parameters based on continuous monitoring of metabolic demands. In future applications, such neuromorphic systems incorporating additional chemosensitive neural populations could be integrated into surgical protocols for intubated patients requiring mechanical ventilation, providing closed-loop neural regulation. Alternatively, these artificial respiratory controllers could be adapted to enhance ECMO systems by enabling more physiological regulation based on continuous feedback from PaCO_2_ sensors. Looking ahead, the integration of physiological monitoring sensors with neuromorphic CPG models could enable the development of fully autonomous systems capable of real-time mapping and regulation of ventilation in response to dynamic physiological states. Ultimately, the next generation implantable devices could incorporate an artificial respiratory CPG built on low-power biocompatible [58,61] artificial hardware [103], offering a potential therapeutic strategy to restore respiratory control in patients with brainstem dysfunction resulting from neural injury or neurodegenerative disease.

## Figures and Tables

**Figure 1 bioengineering-12-01163-f001:**
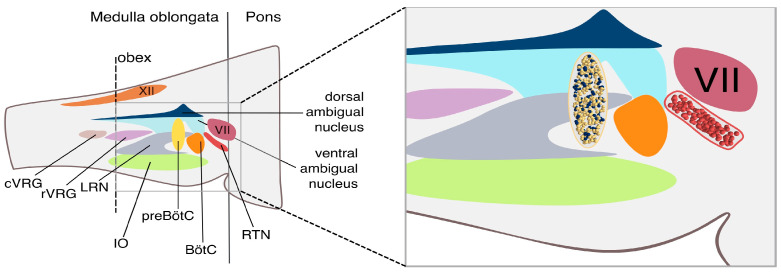
The respiratory CPG circuit. (**Left**). Transversal section of the brainstem illustrating the location of the obex, with clear subdivision into the pons and medulla oblongata. From top to bottom: hypoglossal nerve (XII), dorsal ambigual nucleus, ventral ambigual nucleus, facial nerve (VII), caudal Ventral Respiratory Group (cVRG), rostral Ventral Respiratory Group (rVRG), Lateral Reticular Nucleus (LRN), Inferior Olive (IO), pre-Botzinger complex (preBötC), Botzinger complex (BötC), Retrotrapezoid Nucleus (RTN). (**Right**). Zoomed-in view of the preBötC and RTN neuronal networks at single-cell resolution. The yellow and blue spheres represent individual neurons from the Rhythm and Pattern populations within the preBötC, while the red spheres depict individual¨ neurons located in the RTN.

**Figure 2 bioengineering-12-01163-f002:**
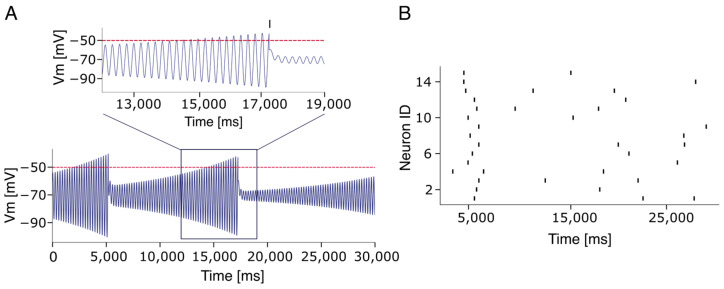
The E-GLIF neuron model. (**A**) Voltage membrane potential of a single E-GLIF optimized to reproduce subthreshold oscillation and an autonomous firing frequency at 0.08 Hz. The inset shows the subthreshold oscillations. Note that spikes are notC strictly evoked when the potential reaches the threshold (dashed red line), but its generation follows a stochastic process determined by the escape rate parameters (λ_0_, τ_v_). (**B**) Firing rate activity of a subpopulation of non-connected EGLIF neurons. Each neuron fires autonomously and independently from the others.

**Figure 3 bioengineering-12-01163-f003:**
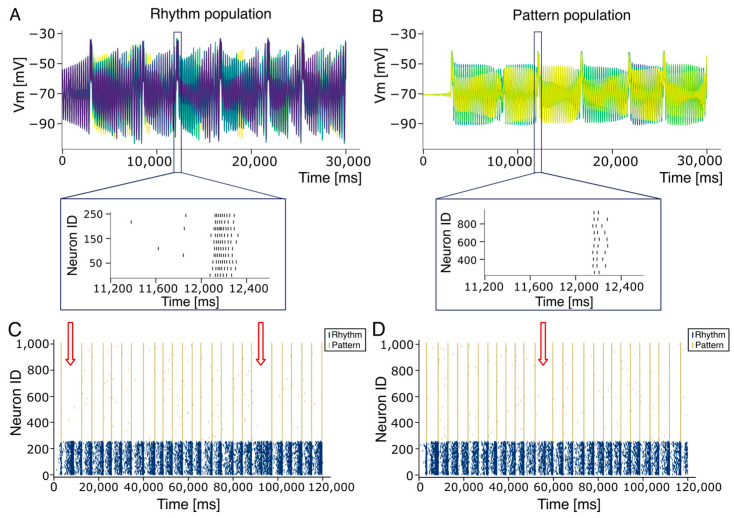
Burst–burstlet synchronization. (**A**) Voltage membrane potentials of a subset (10 traces) of 250 neurons within the rhythm population. The inset shows a zoomed-in view of the bursting activity in the form of a raster plot of emitted spikes by all 250 neurons. (**B**) Voltage membrane potentials of a subset (10 traces) of the 750 neurons within the pattern populations. The inset shows a zoomed-in view of the bursting activity in the form of the raster plot of emitted spike times. Note the reduced intraburst firing frequency in the pattern compared to the rhythm population (22.0 ± 0.3 Hz vs. 50.0 ± 0.5 Hz). (**C**) Raster plot of the Rhythm (blue) and Pattern (yellow) population spike times. The bursting activity is stable in time while showing a little variability in bursting occurrence. Note that a burstlet activation in the Rhythm population is not enough to elicit bursting behaviors in the Pattern Population (red arrows). (**D**) Similar to C but with a different random seed.

**Figure 4 bioengineering-12-01163-f004:**
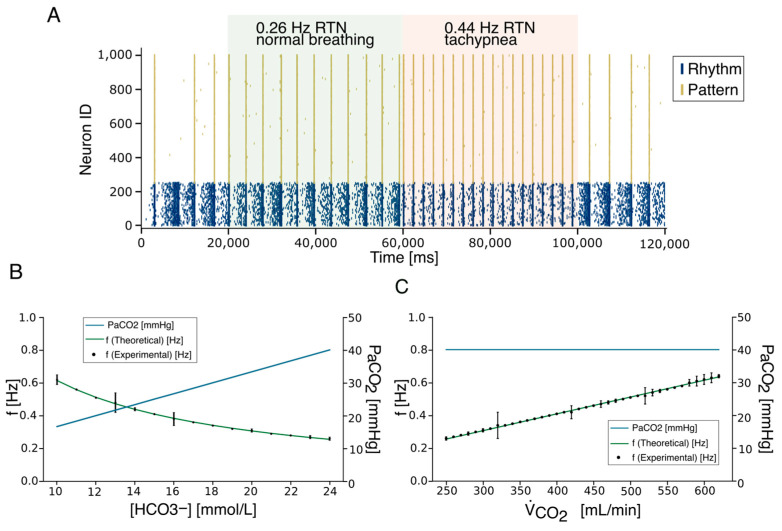
The CPG adaptation to RTN inputs. (**A**) The frequency activity changes in the RTN (green and red boxes) generate the modulation of bursting activity in the rhythm population that is effectively conveyed to the Pattern Population. Two different frequency RTN activities are shown (0.26 Hz and 0.44 Hz). At the end of the RTN activity, the system recovers bursting at physiological frequency (0.22 Hz). (**B**) The respiratory frequency (f, green dashed line) modulation according to the Henderson–Hasselbalch equations. Given a constant value of pH (7.4, purple dashed line), to compensate for a metabolic acidosis perturbation associated with a [HCO3−] reduction, the RR increases while the PaCO_2_ (blue line) decreases. (**C**). The respiratory frequency (f, green dashed line) increases to compensate for variations in carbon dioxide production ($\dot{V}{CO}_{2}$) at constant PaCO_2_ (40.1, blue dashed line). Panels (**B**,**C**) show the experimental frequency values (black dots) with the associated standard error of the mean burst–burstlet synchronization (SEM).

**Figure 5 bioengineering-12-01163-f005:**
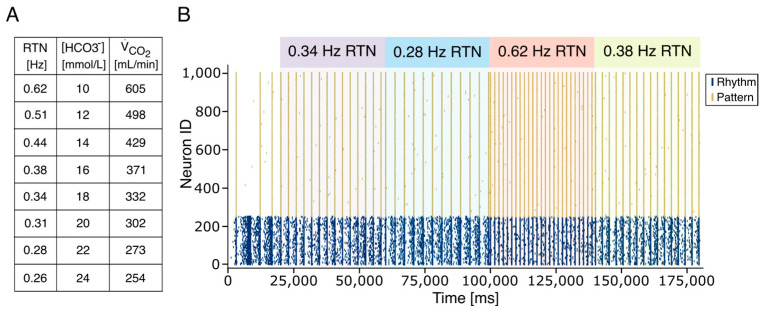
Rhythmogenesis adaptation to multiple physiological changes. (**A**) The table reports the value of the simulated RTN frequencies corresponding to changes in HCO_3_^−^ concentration and $\dot{V}{CO}_{2}$. The perturbation corresponding to a metabolic acidosis (reduction in HCO_3_^−^ concentration) is compensated by an increased RTN burst frequency. (**B**) Physiological accommodation of the rhythmogenesis in the PreBötC is correlated to continuous changes in RTN firing frequency induced by alterations of [HCO_3_^−^] or VCO_2._

**Figure 6 bioengineering-12-01163-f006:**
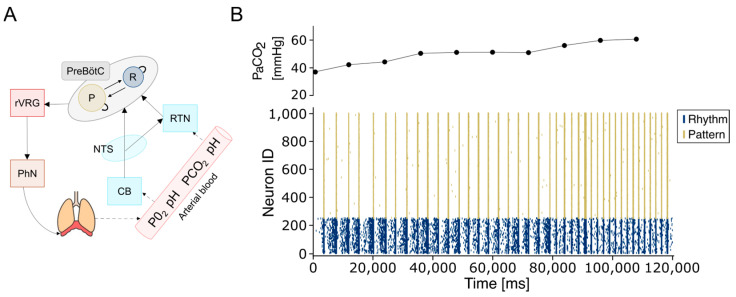
The artificial respiratory CPG. (**A**) Schematic representation of the connections between the PreBötC and the main modulatory and motor areas. The RTN and the carotid bodies detect changes in pH and PaCO_2_ and PaO_2_ values, respectively, and convey these modulatory inputs to the PreBötC. In turn, the PreBötC projects to the ventral respiratory group (VRG), which transmits this rhythmic drive to the phrenic nerve to initiate inspiration. (**B**) Plot illustrating dynamic regulation of breathing in response to metabolic needs. PaCO_2_ is measured every 3.0 s and determines the RTN frequency required to adjust the breathing rate. The upper panel shows PaCO_2_ values every 12 s.

**Table 1 bioengineering-12-01163-t001:** Model parameters adopted to simulate the PreBötC and modified from the default E-GLIF model as described in [40].

	V_m_ (mV)	τ_m_	λ_0_ (ms^−1^)	K_2_ (ms^−1^)	K_adap_ (MH^−1^)	K_2_ (ms^−1^)	A_2_ (pA)	t_ref_ (ms)
Rhythm	−60	1.7	0.0001	0.0331	0.53	0.007	5.0	2.0
Pattern	−70.6	0.9	0.0001	0.0333	0.53	0.007	5.0	2.0

**Table 2 bioengineering-12-01163-t002:** Connection probabilities and synaptic weights between different subpopulations in the modeled network. In bracket connection probability and synaptic weight (p, w):

		Target	
		Rhythm	Pattern
Source	Rhythm	(13%, 9.4)	(30%, 1.0)
	Pattern	(30%, 0.1)	(2%, 0.5)
	RTN	(17%, 2.9)	none

**Table 3 bioengineering-12-01163-t003:** Table lists the parameters with their physiological values. The third and fourth columns report the variation ranges considered in our study.

Parameter	Physiological Value	$\dot{V}{CO}_{2}$—Variation [250–620] mL/min	[HCO_3_^−^] Variation [10–24] mmol/L
pH	7.4	7.4	7.4
[HCO_3_^−^] (mmol/L)	24	24	[10–24]
PaCO_2_ (mmHg)	40.1	40.1	[16.71–40.1]
$\dot{V}{CO}_{2}\;($mL/min)	250	[250–620]	250
${\dot{V}}_{A}\;$*(*L/min)	5.38	[5.38–13.3]	[12.91–5.38]
*V_A_ * (L)	0.35	0.35	0.35
*Bpm* (1/min)	15.4	[15.4–38.1]	[36.9–15.4]
*Interval* (ms)	3903	[3903–1574]	[1626–3903]
*RTN frequency* (Hz)	0.26	[0.26–0.64]	[0.61–0.26]

**Table 4 bioengineering-12-01163-t004:** Comparison of model outputs and physiological data.

	Experimental Data	Model Output	PaCO_2_ Range
Respiratory Rate	(12–35) bpm	(13–40) bpm	all data
Chemoreflex slope (ΔRR/ΔPaCO_2_)	0.7 bpm/mmHg	0.7 bpm/mmHg	(44.3–52.2) bpm/mmHg
Chemoreflex slope (ΔRR/ΔPaCO_2_)	1.5 bpm/mmHg	1.5 bpm/mmHg	(52.2–61) bpm/mmHg

## Data Availability

The original contributions presented in the study are included in the article/Appendix A, further inquiries can be directed to the corresponding authors.

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
