# Peer review of "A Computational Model of the Respiratory CPG for the Artificial Control of Breathing"

_bioengineering, 2025, doi:10.3390/bioengineering12111163_

Round 1
Reviewer 1 Report
Comments and Suggestions for Authors
The manuscript presents a computational model of the respiratory central pattern generator, integrating PreBötC and RTN dynamics for artificial breathing control. The study addresses a timely and relevant topic at the intersection of computational neuroscience, bioengineering, and clinical applications (ECMO and artificial ventilation). While the manuscript is ambitious and demonstrates technical modeling skills, it suffers from several shortcomings in terms of methodological rigor, validation, and clinical translatability.
Major Concerns:
1. The work heavily builds on previous computational models of the PreBötC and burstlet theory. The authors claim novelty by integrating RTN-driven chemoreception and real-time sensor input, but this conceptual advance is incremental rather than groundbreaking. The manuscript should more clearly differentiate what is fundamentally new compared to existing models (e.g., Phillips et al. 2022, 2024; Ashhad et al. 2020).
2. The PreBötC and RTN are modeled with simplified excitatory-only populations, omitting inhibitory and electrical synapses that are known to contribute to synchronization and burst regularization. This abstraction undermines the biological plausibility. The justification for excluding inhibition is insufficient. RTN modeling is overly simplistic (3 spikes at 100 Hz repeated at variable frequency). No validation is provided against experimental firing patterns of RTN neurons under hypercapnia/acidosis.
3. There is no in vitro, in vivo, or patient-derived validation of the model. All claims about clinical applicability (ECMO, intelligent weaning, artificial ventilation) remain speculative. At minimum, the model’s outputs should be compared with published respiratory data (frequency, variability, chemoreflex responses).
4. Many parameters (e.g., connection probabilities, synaptic weights) are “adjusted to obtain desired behavior”, which raises concerns of overfitting and weakens predictive value.
5. The claim that this model “paves the way for clinical closed-loop controllers” is premature. No direct demonstration of safety, robustness, or translational feasibility is shown. A stronger link between simulation outcomes and realistic clinical scenarios is needed.
Reviewer 2 Report
Comments and Suggestions for Authors
The idea of modeling the respiratory CPG for artificial breathing control is sound and promising. The manuscript provides a comprehensive computational framework and integrates it with physiological parameters, which could have significant translational impact. However, several major points need to be carefully considered and addressed before publication.
1.The manuscript describes the E-GLIF model and synaptic parameters in detail, but the description of the simulation protocols, initialization seeds, and parameter tuning still leaves ambiguities. To ensure reproducibility, the authors should provide full parameter tables, code availability, and more explicit details on how key values (e.g., connectivity probabilities, synaptic weights) were derived from experimental data .
2.While the model assumes a conserved PreBötC/RTN architecture across species, the justification for scaling rodent data to humans needs to be strengthened. The absence of direct human data on RTN should be clearly acknowledged, and the limitations of this extrapolation explicitly discussed in the main text, not only in passing in the discussion .
3. The results section convincingly shows burst and burstlet dynamics, but stronger validation against known experimental datasets is needed. For example, frequency responses to hypercapnia or synchronization failures should be quantitatively compared with published in-vivo recordings, not only qualitatively described. This would bolster confidence in the biological plausibility of the model.
4. The manuscript highlights potential applications for ECMO and closed-loop ventilation, which are indeed exciting. However, the limitations for clinical translation (e.g., real-time constraints, patient variability, safety issues) need to be more explicitly stated. The discussion should include what steps are required before moving from simulation to clinical implementation.
5.The Introduction would be significantly improved by situating the proposed respiratory CPG model within the broader trend of AI-driven physiological signal analysis. Beyond respiration, recent work has shown that machine learning and computational methods are increasingly used to detect and interpret neural and behavioral patterns from complex biosignals. To reflect this relevance, I strongly recommend including the following papers in the Introduction:
-
(2025). Automatic calculation of average power in electroencephalography signals for enhanced detection of brain activity and behavioral patterns. Biosensors.
-
(2025). Efficient detection of mind wandering during reading aloud using blinks, pitch frequency, and reading rate. AI.
These studies demonstrate how advanced computational frameworks are already transforming other domains such as EEG analysis and cognitive monitoring. Referencing them will highlight the contemporary relevance of your modeling approach, while also positioning this work as part of a larger movement toward AI-enhanced biomedical and cognitive technologies.
Round 2
Reviewer 1 Report
Comments and Suggestions for Authors
The authors properly reflected my previous comments.
Reviewer 2 Report
Comments and Suggestions for Authors
The new version can be published